# Validation of Psychosocial Measures Assessing American Indian Parental Beliefs Related to Control over Their Children’s Oral Health

**DOI:** 10.3390/ijerph17020403

**Published:** 2020-01-08

**Authors:** Anne R. Wilson, Tamanna Tiwari, Jacob F. Thomas, William G. Henderson, Patricia A. Braun, Judith Albino

**Affiliations:** 1School of Dental Medicine, University of Colorado Anschutz Medical Campus, 13123 E. 16th Ave. B240, Aurora, CO 80045, USA; tamanna.tiwari@cuanschutz.edu; 2Adult and Child Consortium of Outcomes Research and Dissemination Science University of Colorado, Anschutz Medical Campus, 13199 E. Montview Blvd., Suite 300 F443, Aurora, CO 80045, USA; jacob.thomas@cuanschutz.edu (J.F.T.); william.henderson@cuanschutz.edu (W.G.H.); Patricia.braun@cuanschutz.edu (P.A.B.); 3Colorado School of Public Health, University of Colorado Anschutz Medical Campus, 13199 E. Montview, Blvd, Suite 300, W359-G, Aurora, CO 80045, USA; Judith.albino@cuanschutz.edu

**Keywords:** American Indians, dental caries, psychosocial factors, child, validation studies, parents, locus of control

## Abstract

Objectives: To validate questionnaire items assessing American Indian (AI) parental beliefs regarding control over their children’s oral health within the context of psychosocial measures and children’s oral health status. Methods: Baseline questionnaire data were collected as part of a randomized controlled trial (*n* = 1016) addressing early childhood caries. Participants were AI parents with preschool-age children in the Navajo Nation Head Start program. Questionnaire items assessed parental oral health locus of control (OHLOC) and agreement with beliefs indicating that they were in control of their children’s oral health (internal), the dentist was in control (external powerful others), or children’s oral health was a matter of chance (external chance). Exploratory factor analysis was conducted, and convergent validity was assessed using linear regression. Results: Parents with more education (*p* < 0.0001) and income (*p* = 0.001) had higher scores for internal OHLOC. Higher internal OHLOC scores were associated with higher scores on knowledge (*p* < 0.0001), perceived seriousness and benefits (*p* < 0.0001), higher self-efficacy, importance, sense of coherence (*p* < 0.0001 for all), and lower scores for perceived barriers (*p* < 0.0001) and distress (*p* = 0.01). Higher scores for both types of external OHLOC were associated with lower scores on knowledge (*p* < 0.0001), perceived seriousness (*p* < 0.0001), and higher scores on perceived susceptibility (*p* = 0.01 external chance; <0.0001 powerful others) and barriers (<0.0001). Higher scores for external powerful others were associated with lower scores for importance (*p* = 0.04) and sense of coherence (*p* = 0.03). Significant associations were not found for OHLOC beliefs and children’s oral health status. Conclusions: Questionnaire items addressing OHLOC functioned in accordance with the theoretical framework in AI participants.

## 1. Introduction

American Indians and Alaska Natives (AI/AN) experience a higher prevalence of oral health disparities compared to other racial and ethnic groups. On the Navajo Nation Reservation, dental disease is especially severe in children, with 86% of children aged 2–5 years experiencing childhood caries, compared to 62% in AI/AN children overall, and 23% in White children [1,2,3]. As the largest Reservation in the United States, the Navajo Nation encompasses more than 27,000 square miles extending into New Mexico, Arizona, Utah and borders Colorado [4]. Approximately 156,823 individuals reside on the reservation [4], often in small, isolated communities with limited access to public transportation and health care services. Dental care is provided by Indian Health Service with a dentist to population ratio of 32.3 dentists per 100,000 and at the lowest level nationally [5]. In tribal communities, multi-level barriers involving social-structural limitations and behavioral health disparities have contributed to adverse oral health outcomes [6,7]. Understanding the etiology of oral health disparities involving the child–family unit and their environment [7] merits assessment of psychosocial determinants using valid conceptual frameworks [8]. However, few studies have been conducted to assess Native parental influences on children’s oral health outcomes. Thus, using baseline data from a randomized controlled trial aimed at reducing childhood caries among AI children [5], this study assessed the validity of an instrument developed for vulnerable populations at risk for poor oral health to further clarify parental constructs associated with pediatric oral health. 

Integration of theoretical behavioral models within the context of general health has been undertaken to understand an individual’s influence on their health outcomes. The locus of control (LOC) model introduced by Julian Rotter in 1965 and has been broadly applied as an explanatory behavioral model. The LOC model is based on the principle that choice and perceived control resides internally within an individual, or externally with others or the situation [9,10]. The explanatory model assumes behavior follows distinct orientations designated as internal and external LOC, which reflect an individual’s perceived beliefs regarding the connection between their behavior and consequences in a problem-solving context [9,10]. Internal LOC is based on an individual’s belief that there is a connection between their own behavior and outcomes. External LOC is based on an individual’s belief that outcomes are not connected with their own behavior and determined by one of two subscales, chance or powerful others.

Health LOC is a theoretical construct emanating from general locus of control [9,10] and offers ability to inform understanding regarding health-related behaviors, outcomes and care [11]. Health LOC has been widely applied to health research and interventions to measure individual attitudes related to behavioral patterns associated with health and disease. Internal LOC is assigned to individuals with a belief that there is a connection between their own behavior and health outcomes. Within external LOC, the chance subscale is assigned to individuals with a belief that health outcomes are a function of fate, luck or chance while the powerful others is assigned to individuals with a belief that health outcomes are a function of others considered to be authorities such as a physician or dentist. Copious research has identified external LOC as a major determinant of adverse health outcomes [12]. 

As a health behavioral theory, LOC has been studied over many decades. Yet application of LOC in relation to parental influences and children’s oral health outcomes has not been widely studied and results have varied. Among parents of children requiring treatment for early childhood caries utilizing general anesthesia, LOC was not associated with relapse of disease, although parents with higher internal LOC returned for follow-up care [13]. In other studies, internal parental LOC was associated with better control of untreated caries and caries experience in children [14] and mothers with external LOC had children with increased risk of dental caries [15]. Further study and validation of LOC measures relative to parental influences and children’s oral health outcomes is merited. The purpose of this study was to examine the validity of oral health surveillance items developed to assess parental LOC within the context of other psychosocial measures (oral health knowledge and behavior, health belief model, self-efficacy, importance, distress, sense of coherence) and oral health status of children and parents. Based on the theoretical model for health LOC, construct validity is expected for the OHLOC items used as measures of parental factors influencing children’s oral health outcomes in an AI cohort.

## 2. Materials and Methods

### 2.1. Study Approvals

This study was approved by the Navajo Nation Human Research Review Board, governing bodies at tribal and local levels, tribal departments of Head Start and Education, Head Start parent councils, and the Multiple Institutional Review Board of the University of Colorado. Anschutz Medical Campus, Aurora, Colorado. The study protocol (08-0892) was approved on 28 August 2009 by the Multiple Institutional Review Board of the University of Colorado. 

### 2.2. Study Design

The study was described in earlier reports [16,17], and only the key features will be presented here. The study was a cluster-randomized trial, with randomization at the level of the Navajo Nation Head Start Center. The Head Start Centers were stratified by agency (region of the reservation) and whether the Head Start Center had one or multiple classrooms and then randomized to an intervention to prevent early childhood caries or usual care. The final sample included 39 Head Start Centers (19 control and 20 intervention Head Start Centers), with 26 classrooms per group. Participants were recruited as parent/caregiver–child dyads. Children were eligible if they were three to five years old, enrolled in a participating Head Start Center, and their parent/caregiver provided informed consent to participate in the study (hereafter referred to as “parents”). All adult subjects gave their informed consent for inclusion before they participated in the study. The study was conducted in accordance with the rules of the Declaration of Helsinki of 1975 and the 2013 revision. 

To understand parental influences on the oral health of their children, parents completed the Basic Research Factors Questionnaire (BRFQ) [18] at baseline and annually for up to three years. The BRFQ is a 190 item questionnaire encompassing sociodemographic characteristics for the children and parents, parental oral health knowledge and behaviors, and parental oral health attitudes relative to psychosocial measures. Many of the psychosocial measures from the BRFQ have been evaluated as part of validation studies [19,20,21,22] excluding the LOC measures. In addition, oral health status for children and parents was evaluated at baseline and annually up to three years. For the purposes of this study, only baseline data were used for the validation assessment of the LOC measures. 

### 2.3. Measures

Construct validity, specifically convergent and divergent validity, determined the relatedness among measures. Convergent validity determined the degree to which two measures expected to be related are, indeed, related. Divergent validity measured the degree to which two measures expected to be unrelated are, indeed, unrelated.

### 2.4. Oral Health Locus of Control

The LOC construct used in this study is referred to as oral health locus of control (OHLOC) and adapted from the original construct as the primary focus of this analysis. Nine items assessed OHLOC, which measured the parents’ beliefs regarding the source of control over their children’s oral health. Items (b–i) were adapted from existing measures [14] and the work of TM Carnahan, “The development and validation of the multidimensional dental locus of control scales” (unpublished doctoral dissertation, State University of New York at Buffalo; 1979). Item a. was newly developed to provide a second item in the external-powerful other subscale (Table 1). Items determined the extent to which parents agreed with statements indicating that they were in control of their children’s oral health reflecting an internal control belief (OHLOC-I), their children’s oral health was a matter of chance reflecting an external chance belief (OHLOC-EC) or the dentist was in control reflecting an external powerful others belief (OHLOC-EO). Items used a scale of one to five, where one represented “strongly disagree” and five represented “strongly agree”. For each type of OHLOC, the average of three items assessing that domain was computed. Larger numbers for each subscale represented endorsement of that aspect of OHLOC.

The exploratory factor analysis yielded two factors, accounting for 52% of the total variance of the nine OHLOC questions. The two factors were internal and external OHLOC. Both types of external OHLOC, (chance and powerful others) tended to load on the same factor (Table 1). 

Significant relationships resulted for several sociodemographic variables and the OHLOC subscales (Table 3). Parents with more education (*p* < 0.0001) and household income (*p* = 0.001) had higher scores on the OHLOC-I subscale. Female (*p* = 0.01) and older (*p* = 0.004; OHLOC-EC and *p* = 0.01; OHLOC-EO) parents, and those with more education and income (*p* < 0.0001) tended to have lower scores for both OHLOC-EC and OHLOC-EO. 

### 2.5. Oral Health Knowledge and Behavior

Fourteen questions examined parents’ oral health knowledge and 11 questions examined adherence to recommended oral health behaviors in taking care of their children’s teeth. For oral health knowledge, responses were coded as correct or incorrect (“don’t know” responses were coded as incorrect). An overall knowledge score was computed as a percentage of questions answered correctly. Responses to oral health behavioral recommendations were coded as adherent or non-adherent. The overall behavioral adherence score was the percentage of behaviors for which the parents were adherent to oral health recommendations.

### 2.6. Health Belief Model

Sixteen items measured four key constructs of the health belief model [23,24]. Key constructs included perceived susceptibility (parents’ perceptions that their children were susceptible to cavities), perceived seriousness (degree to which parents believed oral health problems were serious), and perceived benefits and barriers to engaging in recommended oral health behaviors for their children. Items were adapted from four sources [25,26,27,28]. Responses to all items ranged from one (strongly disagree) to five (strongly agree). The average of the items associated with each construct was computed. Larger numbers represented a greater degree of each construct.

### 2.7. Self-Efficacy

Twelve items were used to measure self-efficacy, a key construct from social cognitive theory [29,30] representing an individual’s confidence in successfully engaging in recommended health behaviors. Some items were adapted from Reisine’s dental confidence questionnaire [31], and others newly developed. All items asked parents to indicate their confidence level in successfully engaging in recommended oral health behaviors. Items used a scale of one to five, ranging from “not at all sure” to “extremely sure”. The average of the self-efficacy items was computed for each participant. Larger numbers represented a greater degree of self-efficacy.

### 2.8. Importance

Twelve importance items were identical to the self-efficacy items, except that parents indicated how important it was to engage in each oral health behavior. Items used a scale of one to five, ranging from “not at all important” to “extremely important”. For each participant, the average of the importance items was computed.

### 2.9. Distress

We used the 6 item K6 nonspecific psychological distress scale developed by Kessler, et al. [32] to measure parent distress. Responses were measured on a scale of one to five, with higher scores indicating more distress.

### 2.10. Sense of Coherence

We used the 13 item short-form sense of coherence scale [33]. Responses for all items ranged from one to seven. Sense of coherence measures the degree to which the individual views the world and her or his life circumstances as coherent, an orientation that may support constructive responses to challenging life events, including a variety of health problems. For each participant, we computed the average of the 13 items.

### 2.11. Divergent Measures

To demonstrate that positive associations for convergent measures were not spurious, BRFQ items expected to be unrelated to oral health locus of control were selected as divergent validity measures: baseline survey year (2011 or 2012); whether the Head Start Center had single or multiple classrooms; and agency where the Head Start Center was located (coded one to five). For the divergent validity measures, the health LOC measures were not expected to change or be different for the various years of the survey, the Head Start Centers with single or multiple classrooms, or the various agencies within the Navajo Nation.

### 2.12. Indicators of Oral Health Status

Three measures were used as indicators of children’s oral health status; a single measure assessed parent oral health. The three child measures included a count of decayed, missing, or filled tooth surfaces (dmfs) [34]; an item adapted from the National Survey of Children’s Health [35], asking parents to rate their children’s oral health status as 1 = excellent, 2 = very good, 3 = good, 4 = fair, or 5 = poor; and a pediatric oral health quality of life measure (POQL), ranging from 0 = best to 100 = worst [19,36]. The parent oral health status measure asked the parent to rate their own oral health status on a scale of 1 = excellent to 5 = poor.

### 2.13. Data Analysis

Descriptive statistics (means, standard deviations, and percentages) were used to characterize the sociodemographic variables and the baseline psychosocial measures for the sample of parents and children.

An exploratory factor analysis was conducted to determine the underlying structure of the OHLOC questions. The factor analysis program was allowed to choose the number of factors. The exploratory factor analysis tends to group together those OHLOC questions with the highest correlations among themselves.

The association between the OHLOC subscales and the sociodemographic characteristics of the parents and the divergent measures was examined by comparing the mean OHLOC subscales across the categories of the sociodemographic variables and the divergent measures. Differences in mean OHLOC subscale values were compared using a one-factor analysis of variance.

The association between the OHLOC subscales and the other psychosocial measures was determined using simple linear regression analysis—in which, the dependent variable was the psychosocial measure and the independent variable was the OHLOC subscale score. The regression analyses were adjusted for differences in the sociodemographic variables of the parents. The alpha level for statistical significance was set at 0.05. For all analyses, SAS, version 9.4, (SAS Institute, Cary, NC, USA) was used.

## 3. Results

Table 2 presents the sociodemographic characteristics of the parent and child sample and the mean baseline scores for the psychosocial measures of the parents. Approximately 84% of the parents were female, with a mean age of 32 years. Forty-seven percent of the parents had some college or a college degree. However, mean household annual income was low, with 59% under $20,000. The parents generally had high scores (over 4 on a 5-point scale) for internal OHLOC, perceived seriousness and benefits from the health belief model, self-efficacy, importance, and sense of coherence. 

The exploratory factor analysis yielded two factors, accounting for 52% of the total variance of the nine OHLOC questions. The two factors were internal and external OHLOC. Both types of external OHLOC, (chance and powerful others) tended to load on the same factor (Table 1). 

Significant relationships resulted for several sociodemographic variables and the OHLOC subscales (Table 3). Parents with more education (*p* < 0.0001) and household income (*p* = 0.001) had higher scores on the OHLOC-I subscale. Female (*p* = 0.01) and older (*p* = 0.004; OHLOC-EC and *p* = 0.01; OHLOC-EO) parents, and those with more education and income (*p* < 0.0001) tended to have lower scores for both OHLOC-EC and OHLOC-EO. 

Many of the OHLOC subscales were related to the other psychosocial measures (Table 4). Higher OHLOC-I scores were associated with higher scores on oral health knowledge (*p* < 0.0001) (but not oral health behavior), perceived seriousness and perceived benefit subscales of the health belief model (*p* < 0.0001), and higher self-efficacy, importance, and sense of coherence (*p* < 0.0001 for all), and lower scores for perceived barriers (*p* < 0.0001) and distress (*p* = 0.01). On the other hand, higher scores for both OHLOC-EC and OHLOC-EO were associated with lower scores on oral health knowledge (*p* < 0.0001), perceived seriousness (*p* < 0.0001), and higher scores on perceived susceptibility (*p* = 0.01 and <0.0001) and barriers (<0.0001) of the health belief model. Also, higher scores for OHLOC-EO were associated with lower scores for importance (*p* = 0.04) and sense of coherence (*p* = 0.03).

Regression coefficients are from a simple linear regression analysis—in which, the dependent variable is the psychosocial variable and the independent variable is the OHLOC subscale; therefore, they represent the change in the psychosocial variable per unit increase in the OHLOC subscale.

There were no statistically significant associations between the OHLOC subscales and the chosen divergent measures (Table 5).

Finally, there were relatively few statistically significant associations between the OHLOC subscales and the oral health status outcomes (Table 6). The two that were statistically significant were the associations between parent oral health status and the OHLOC-EC and OHLOC-EO subscales (*p* = 0.002 and *p* = 0.001 respectively). Parents with higher scores for both of the OHLOC external subscales tended to rate their own oral health status as better.

Regression coefficients are from a simple linear regression analysis—in which, the dependent variable is the oral health status measure and the independent variable is the OHLOC subscale; therefore, they represent the change in the oral health status measure per unit increase in the OHLOC subscale.

## 4. Discussion

Together with the LOC theory, a range of behavioral models relative to general health have been proposed to address oral health disparities including sense of coherence [21,37], the health belief model [22], and self-efficacy [22,31]. Within the context of oral health, psychosocial determinants are considered to have a major impact on oral health disparities. Nonetheless, application of theoretical models to provide insight for existing oral health disparities in young children has been minimally studied. This study is the first to investigate and validate oral health measures assessing AI parental beliefs regarding control over their children’s oral health in relation to other psychosocial measures and children’s oral health status. 

Study outcomes confirmed many of the convergent measures were significantly associated with the OHLOC subscales. Consistent with the theoretical basis of the LOC model, parents with an internal orientation or belief that their behavior influences health outcomes, perceived greater benefits to and fewer barriers in adherence with oral health recommendations, perceived dental caries as a serious problem, and had greater confidence in their ability to manage their children’s oral health. Additionally, parents with an internal orientation reported less distress, a higher sense of coherence and potential to constructively respond to health problems and viewed engaging in recommended oral health behavior as highly important. Contrary to expectations, parents with an internal orientation believed their children were susceptible to developing dental caries. Similar findings existed in separate studies evaluating the health belief model in this AI cohort [22]. These findings may be explained by generational persistence of extreme oral disease in AI children and adults [3]. Within these environments, susceptibility to dental caries may remain a perpetual concern despite having an internal orientation. 

Based on hypothetical expectations (Figure 1), health beliefs were expected to be related to knowledge, behavior, and health outcomes. Based on the directional premise of the model, parental knowledge was predicted to influence beliefs regarding control of their children’s oral health, followed by beliefs contributing to oral health behavior and ultimately children’s oral health outcomes. Study results were consistent with parental oral knowledge associated with all OHLOC subscales, while no association was demonstrated for parental oral health behavior. Previous studies have demonstrated the degree of behavioral adherence was notably lower compared to oral health knowledge [20]. These findings suggest knowledge is a contributory factor, although not sufficient to elicit change in health behavior patterns [20]. Correspondingly, parental OHLOC did not influence children’s oral health status or dmfs scores. 

Many of the sociodemographic characteristics were associated with the OHLOC subscales. As hypothesized, lower scores for an internal orientation and higher scores for an external orientation were associated with reduced income and educational attainment. Research has reflected that individuals having the lowest income and education level have the worst health and oral health outcomes and experience worse disease sooner [8]. Female and older parents were more likely to have lower scores for external powerful others, reflecting their belief that children’s oral health outcomes were connected to their parenting efforts. These findings may be explained by the strong matriarchal influence in tribal communities, with AI women’s identities being closely tied to their roles as mothers, grandmothers, and caregivers [38]. 

The study had limitations that are important to note. The analyses were based on cross-sectional data. Thus, it was not possible to examine the causal direction of relationships to determine whether perceptions influence behavior and outcomes or whether behavior and outcomes influence perceptions. Exploratory factor analysis was conducted to demonstrate the underlying structure of variables. However, other measures of validity, such as face validity, content validity, and concurrent validity, were not part of the validation studies. The instrument is validated for the AI population, with results specific to this population and not the general population of all parents and children.

## 5. Conclusions

In summary, results from this study suggest that the items assessing the OHLOC theoretical constructs were valid as measures of parental factors influencing children’s oral health outcomes in an AI cohort. The convergent relationships between OHLOC and sociodemographic variables and other psychosocial measures were consistent with the hypothetical direction of the model, indicating that the OHLOC measures had strong convergent validity in a cohort of AI parents. Construct validity for the factor model was supported with both types of external LOC loading on the same factor and internal OHLOC loading on another factor. These analyses support the use of the OHLOC surveillance items as validated measures for understanding parental psychosocial factors influencing the oral health status of their children.

## Figures and Tables

**Figure 1 ijerph-17-00403-f001:**
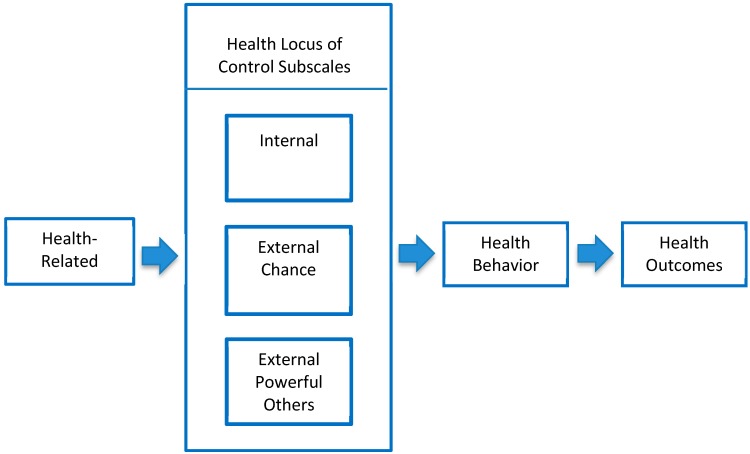
Health locus of control model.

**Table 1 ijerph-17-00403-t001:** Exploratory factor analysis after varimax rotation (N = 802).

	Question	OHLOC Subscale	Factor 1 Loading	Factor 2 Loading
a.	Preventing cavities is in the hands of the dentist.	External-powerful others	0.75731	−0.08461
b.	It’s the dentist’s job to keep my child from getting cavities.	External-powerful others	0.72176	−0.19282
c.	Luck plays a big part in how healthy my child’s teeth are.	External-chance	0.71503	−0.18897
d.	The dentist is the best person to prevent cavities in my child.	External-powerful others	0.70976	0.10632
e.	Having good teeth is largely a matter of luck.	External-chance	0.65678	0.16045
f.	Some children just naturally have soft teeth.	External-chance	0.59882	0.15218
g.	It is up to me to make sure my child doesn’t get cavities.	Internal	0.00564	0.83748
h.	I am sure that I can reduce the chances of my child getting cavities.	Internal	0.02474	0.80429
i.	If my child gets cavities, I am to blame.	Internal	−0.01404	0.52179
	Percent of total variance explained by each factor		32%	20%

OHLOC—oral health locus of control.

**Table 2 ijerph-17-00403-t002:** Sample characteristics (N = 1016).

Parent Characteristics	N	Mean (SD) or %
Age	1016	31.9 (9.3)
Gender: Female	851	83.8%
Highest Grade Completed		
<High school graduate	159	15.8%
High school grad/GED	373	37.2%
Some college/vocational	353	35.2%
College degree or more	119	11.9%
Income		
<$10K	422	41.5%
$10K to <$20K	176	17.3%
$20K to <$30K	94	9.3%
$30K to <$40K	69	6.8%
≥$40K	91	9.0%
Income missing	164	16.1%
Internal Locus of Control	1006	4 (1)
External Locus of Control-Chance	1004	2.5 (1.1)
External Locus of Control-Powerful Others	1003	2.3 (1.1)
Oral Health Knowledge Score (%-Correct)	1010	74.3 (13.4)
Oral Health Behavior Score (%-Adherent)	1010	50.8 (22.2)
Perceived Susceptibility	1004	3.4 (0.9)
Perceived Seriousness	1007	4.3 (0.8)
Perceived Barriers	1009	2.2 (0.7)
Perceived Benefits	1007	4.3 (0.8)
Self-Efficacy	1010	4.4 (0.5)
Importance	1009	4.7 (0.4)
Distress	1002	1.7 (0.7)
Sense of Coherence	986	5.2 (1.1)
Child Characteristics		
Age	1014	3.6 (0.5)
Gender: Female	517	50.90%

**Table 3 ijerph-17-00403-t003:** Relationship between parent characteristics and OHLOC subscales.

		Internal (OHLOC-I)		External Chance (OHLOC-EC)		External Powerful Others (OHLOC-EO)	
Variable	Response	Mean (SD)	*p*-Value	Mean (SD)	*p*-Value	Mean (SD)	*p*-Value
Gender	Male (*n* = 162)	4.1 (0.9)	0.13	2.3 (1.1)	0.10	2.5 (1.1)	0.01
	Female (*n* = 841)	4.0 (1.0)		2.5 (1.1)		2.2 (1.1)	
Age	19–25 (*n* = 276)	4.0 (1.0)	0.054	2.7 (1.1)	0.004	2.4 (1.1)	0.01
	26–30 (*n* = 274)	4.0 (0.9)		2.4 (1.0)		2.2 (1.0)	
	31–36 (*n* = 217)	4.2 (0.9)		2.3 (1.2)		2.1 (1.1)	
	37–88 (*n* = 236)	3.9 (1.0)		2.5 (1.2)		2.3 (1.1)	
Education	<HS (*n* = 155)	3.8 (1.1)	<0.0001	2.8 (1.1)	<0.0001	2.5 (1.1)	<0.0001
	HS/GED (*n* = 372)	4.0 (1.0)		2.7 (1.2)		2.4 (1.1)	
	Some college (*n* = 353)	4.2 (0.9)		2.2 (0.9)		2.0 (0.9)	
	College degree (*n* = 118)	4.2 (0.9)		2.1 (0.9)		1.9 (0.9)	
Income	<$10K (*n* = 419)	3.9 (1.0)	0.001	2.7 (1.1)	<0.0001	2.5 (1.1)	<0.0001
	$10K < $20K (*n* = 176)	4.1 (0.9)		2.2 (0.9)		2.0 (0.8)	
	$20K < $30K (*n* = 94)	4.3 (0.8)		2.2 (1.0)		2.1 (1.0)	
	$30K < $40K (*n* = 69)	4.1 (0.9)		2.1 (0.9)		2.0 (0.8)	
	≥$40K (*n* = 91)	4.3 (0.7)		2.0 (1.0)		1.9 (1.0)	
	Missing (*n* = 153)	3.9 (1.1)		2.6 (1.2)		2.4 (1.2)	
Employment	Employed (*n* = 279)	4.1 (0.9)	0.13	2.3 (1.0)	0.051	2.2 (1.0)	0.15
	Other (*n* = 702)	4.0 (1.0)		2.5 (1.1)		2.3 (1.1)	

OHLOC—oral health locus of control; OHLOC-I (internal); OHLOC-EC (external chance); OHLOC-EO (external powerful others); HS—high school.

**Table 4 ijerph-17-00403-t004:** Convergent validity—association between OHLOC subscales and other psychosocial measures (N range = 978–1000).

	Internal (OHLOC-I)		External Chance (OHLOC-EC)		External Powerful Others (OHLOC-EO)	
Psychosocial Measure	Regression Coefficient	*p*-Value	Regression Coefficient	*p*-Value	Regression Coefficient	*p*-Value
Oral health knowledge score	1.92	<0.0001	−2.14	<0.0001	−2.61	<0.0001
Oral health behavior score	0.22	0.77	−0.22	0.75	-0.73	0.29
Health belief model						
Perceived susceptibility	−0.004	0.88	0.07	0.01	0.11	<0.0001
Perceived seriousness	0.24	<0.0001	−0.17	<0.0001	−0.22	<0.0001
Perceived barriers	−0.15	<0.0001	0.10	<0.0001	0.13	<0.0001
Perceived benefits	0.19	<0.0001	−0.01	0.73	0.02	0.35
Self-efficacy	0.11	<0.0001	0.01	0.41	−0.02	0.35
Importance of oral health behaviors	0.09	<0.0001	-0.01	0.32	−0.03	0.04
Distress	−0.06	0.01	0.01	0.62	0.03	0.22
Sense of coherence overall score	0.15	<0.0001	−0.04	0.26	−0.07	0.03

OHLOC—oral health locus of control; OHLOC-I (internal); OHLOC-EC (external chance); OHLOC-EO (external powerful others).

**Table 5 ijerph-17-00403-t005:** Divergent validity—association between OHLOC subscale and each presumed divergent measure.

	Internal (OHLOC-I)		External Chance (OHLOC-EC)		External Powerful Others (OHLOC-EO)	
Presumed Divergent Measure	Mean (SD)	*p*-Value	Mean (SD)	*p*-Value	Mean (SD)	*p*-Value
Year of Survey		0.56		0.19		0.11
2011 (*n* = 558)	4.0 (0.9)		2.4 (1.1)		2.2 (1.0)	
2012 (*n* = 444)	4.0 (1.0)		2.5 (1.1)		2.3 (1.1)	
Head Start Center Classrooms		0.16		0.30		0.12
Single (*n* = 593)	4.0 (1.0)		2.5 (1.1)		2.3 (1.1)	
Multiple (*n* = 410)	4.1 (0.9)		2.4 (1.1)		2.2 (1.1)	
Navajo Nation Agency		0.91		0.26		0.06
1 (*n* = 235)	4.0 (1.0)		2.6 (1.1)		2.3 (1.1)	
2 (*n* = 173)	4.1 (1.0)		2.4 (1.1)		2.1 (1.0)	
3 (*n* = 198)	4.0 (1.0)		2.5 (1.2)		2.4 (1.1)	
4 (*n* = 189)	4.1 (0.9)		2.5 (1.1)		2.2 (1.1)	
5 (*n* = 206)	4.0 (1.0)		2.3 (1.1)		2.2 (1.0)	

OHLOC—oral health locus of control; OHLOC-I (internal); OHLOC-EC (external chance); OHLOC-EO (external powerful others); SD—standard deviation.

**Table 6 ijerph-17-00403-t006:** Association between OHLOC subscale and oral health status of children and parents (N range = 965–991).

	Internal (OHLOC-I)		External Chance (OHLOC-EC)		External Powerful Others (OHLOC-EO)	
Oral Health Status Measure	Regression Coefficient	*p*-Value	Regression Coefficient	*p*-Value	Regression Coefficient	*p*-Value
Child dmfs	−0.46	0.50	0.88	0.15	0.57	0.36
Child oral health status	−0.02	0.52	−0.04	0.24	−0.002	0.95
Child POQL	−0.16	0.62	0.45	0.13	0.53	0.07
Parent oral health status	0.06	0.06	−0.09	0.002	−0.10	0.001

OHLOC—oral health locus of control; OHLOC-I (internal); OHLOC-EC (external chance); OHLOC-EO (external powerful others); dmfs—decayed, missing, and filled tooth surfaces in primary teeth; POQL—pediatric oral health quality of life.

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
