# Peer review of "Validation of Psychosocial Measures Assessing American Indian Parental Beliefs Related to Control over Their Children’s Oral Health"

_ijerph, 2020, doi:10.3390/ijerph17020403_

Round 1
Reviewer 1 Report
Thank you for the opportunity to review this manuscript. I consider this research very important in the oral epidemiology field. The paper in general is well-written and there exist a coherence among all the sections and the methodology seems so adequate to accomplish the study objectives.
I recommend that the authors check the requirements of the journal and to follow the guidelines to present the figure and the tables according to the template that is purposed for the journal.
Similarly, the authors should consider and review the guidelines to report different studies according to the study design. I know that the study is part of a big project but in this case, it seems an observational study and it is important to review the accomplishment the STROBE guidelines.
Please refer or mention the specific study design.
It is necessary to mention the limitations of the study and the recommendations for research and action (discussion section)
Reviewer 2 Report
Authors presented the results of a validation study of a tool to assess the parental oral health locus of control (OHLOC) and its association with children's oral health outcomes. The LOC tool is part of a 190-item tool (Basic Research Factors Questionnaire (BRFQ)) that was developed in 2009 and used across four Randomized Trials. The results of these studies provided the foundation for four validation studies by now: pediatric oral health-related quality of life scale [1], caregivers’ oral health 380 knowledge and behavior [2] Sense of Coherence Scale[3], and oral health behavior beliefs of American Indian Patients [4]. In this study, investigators present the results of validation study of parental oral health locus of control (OHLOC) items.
Major Comments:
The research justification and the rational of this study is not clear. As authors presented, 8 items out of this 9-item questionnaire was adopted from an existing tool[5] and authors just added one item (item a) to develop this construct. Are they trying to present their tool as a new LOC measure that could be used in other studies? If yes, justification is needed to clarify what distinguishes this tool from existing tool(s). They also need to clarify if they have permission from authors of the existing tool to adopt items. No information regarding the reliability of this tool (Internal consistency, Test-retest reliability, Inter-rater reliability) has been presented. Authors presented results of exploratory factor analysis to show the underlying structure of variables. However, other measures of validity, such as Face validity, content validity, concurrent validity, etc. have not presented. Lines 242-243: authors mentioned " There were no statistically significant associations between the OHLOC subscales and the chosen divergent measures" These measures should be described in more details in methods and results. I would like to see some discussion of the findings of this study, how this tool is different from existing ones and how could be used and administered in future studies.
[1] Braun, P.A.; Lind, K.E.; Henderson, W.G.; Brega, A.G.; Quissell, D.O.; Albino, J. Validation of a pediatric oral health-related quality of life scale in Navajo children. Qual. 377 Life Res. 2015, 24, 231-239.
[2] Wilson, A.R.; Brega, A.G.; Campagna, E.J.; Braun, P.A.; Henderson, W.G.; Bryant, L.L.; Batliner, T.S.; Quissell, D.O.; Albino, J. Validation and impact of caregivers’ oral health 380 knowledge and behavior on children’s oral health status. Pediatr. Dent. 2016, 38(1), 47-54.
[3] Albino, J.; Shapiro, A.L.B.; Henderson, W.G.; Tiwari, T.; Brega, A.G.; Thomas, J.F.; Bryant, L.L.; Braun, P.A.; Quissell, D.O. Validation of the Sense of Coherence scale in an American Indian population. Psychol. Assess. 2016, 28(4), 386-393.
[4] Wilson, A.R.; Brega, A.G.; Thomas, J.F.; Henderson, W.G.; Lind, K.E.; Braun, P.A.; Batliner, T.S.; Albino, J. Validity of measures assessing oral health beliefs of American 386 Indian parents. J. Racial Ethn. Health Disparities 2018, 6, 1254-1263.
[5] Lencová, E.; Pikhart, H.; Broukal, Z.; Tsakos, G. Relationship between parental locus of control and caries experience in preschool children: cross-sectional survey. BMC Public 360 Health 2008, 8, 208.

Round 2
Reviewer 2 Report
Thank you for your responses.
Please add the explanation that you provided as the justification of your study to the introduction section.
Also, as you mentioned, this study has certain limitations that inhibits its generalizability. Please revise your manuscript to reflect that this instrument is validated to be used in American Indian population and results of this study is limited to this group not the general population of all parents and children.
